# Stochastic gene expression and environmental stressors trigger variable somite segmentation phenotypes

Kemal Keseroglu [1], Oriana Q. H. Zinani[1,2], Sevdenur Keskin[3], Hannah Seawall [1], Eslim E. Alpay [1] & Ertuğrul M. Özbudak [1,4] ✉

Mutations of several genes cause incomplete penetrance and variable expressivity of phenotypes, which are usually attributed to modifier genes or gene-environment interactions. Here, we show stochastic gene expression underlies the variability of somite segmentation defects in embryos mutant for segmentation clock genes *her1* or *her7*. Phenotypic strength is further augmented by low temperature and hypoxia. By performing live imaging of the segmentation clock reporters, we further show that groups of cells with higher oscillation amplitudes successfully form somites while those with lower amplitudes fail to do so. In unfavorable environments, the number of cycles with high amplitude oscillations and the number of successful segmentations proportionally decrease. These results suggest that individual oscillation cycles stochastically fail to pass a threshold amplitude, resulting in segmentation defects in mutants. Our quantitative methodology is adaptable to investigate variable phenotypes of mutant genes in different tissues.

Some deleterious gene mutations only affect a subset of genetically mutant animals (i.e., incomplete penetrance) while others affect individuals at different phenotypic strength (i.e., variable expressivity). These widespread phenomena complicate the discovery of causative gene mutations in both model organisms and human diseases. Human individuals harboring mutations for severe Mendelian conditions, thought to be fully penetrant, occasionally do not display a disease phenotype[1]. Thus, incomplete penetrance and variable expressivity of phenotypes among human genetic diseases might be more widespread than previously appreciated. Proposed mechanisms for incomplete penetrance and variable expressivity of human disease-causing alleles include environmental risk factors and additional genetic variations (e.g., modifier genes)[2–6]. However, these mechanisms cannot fully account for phenotypic differences as phenotypic inhomogeneity exists even in the absence of genetic variation, for example between monozygotic twins[7]. Thus, the sources of both incomplete penetrance and variable expressivity of phenotypes

remain to be elucidated. Pioneering studies in *C. elegans* suggested that gene expression variability might be the source of incomplete penetrance in some mutant backgrounds[8,9]. Whether this conclusion can be transferable to a vertebrate system as well as extendable to the variable expressivity of phenotypes in other mutants remained to be shown. Furthermore, phenotypes investigated in *C. elegans* result in binary outcomes (i.e., normal or abnormal development) which can only be compared among different embryos[8,9]. However, metameric and quantitative phenotypes can also be investigated in a single embryo, providing an unmatched advantage for investigating factors that influence phenotypic variability.

Somitogenesis –sequential and periodic segmentation of somites– generates the metameric organization of the major body axis of vertebrates. Somites are the precursors to the musculoskeletal system of the vertebral column. To form the full-length body axis, segmentation continues for a species-specific number of cycles (e.g., 33 cycles in zebrafish). Depending on their axial position, each somite

[1]Division of Developmental Biology, Cincinnati Children's Hospital Medical Center, Cincinnati, OH 45229, USA. [2]Molecular and Developmental Biology Graduate Program, University of Cincinnati, College of Medicine, Cincinnati, OH 45229, USA. [3]Allergy and Immunology, University of Arkansas for Medical Science and Arkansas Children's Hospital, Little Rock, AR 72202, USA. [4]Department of Pediatrics, University of Cincinnati College of Medicine, Cincinnati, OH 45229, USA. ✉e-mail: ertugrul.ozbudak@cchmc.org

in zebrafish contains 100 to 200 cells on average. The period of somitogenesis is controlled by the oscillatory expression of the segmentation clock genes[10]. In zebrafish, two linked genes, *her1* and *her7*, have been identified as central to the genesis of oscillations[11–13] (Fig. 1a). *her1* and *her7* code for bHLH family transcriptional repressors that function as dimers. Her1 binds to DNA as a homodimer while Her7 binds to DNA as a heterodimer with a non-oscillating Hes6 partner[14,15]. Her1-Her1 and Her7-Hes6 repress transcription of *her1* and *her7*, and therein form a negative feedback loop, which is necessary for establishing a limit cycle triggering their oscillations[16,17]. When *her1* and *her7* are jointly mutated, all signs of oscillation are lost, and segment boundaries are defected throughout the entire body axis[11–13]. In contrast, in single loss of function mutants, it has been reported that boundary defects display variable phenotypes. In *her1* mutant embryos, few defected boundaries were detected, usually in the anterior-most somites[15,17,18]. In *her7* mutant embryos, anterior somite boundaries (approximately up to the eighth) were unaffected; defects were confined to the posterior trunk and tail region[15,17,18]. Individual embryos of the same mutant genotype varied in the extent of segmentation defects though the strength of phenotypes have not been investigated in detail[17]. Because cells segmenting into different somites in each embryo are genetically identical, segmentation mutants provide us a unique opportunity to quantitatively investigate the underlying sources of both incomplete penetrance and variable expressivity of segmentation phenotypes in single vertebrate embryos.

In this work, we find that stochastic gene expression is the fundamental source of phenotypic variability for the segmentation defects within mutant embryos. We also find that the phenotypic severity increases under unfavorable environmental factors, such as low temperature and hypoxia. Through live imaging of the segmentation clock reporters, we further show that segmentation success correlates with the collective oscillation amplitude. Our results suggest that stochastic failure of clock oscillations in groups of cells results in variable somite segmentation defects for both *her1* and *her7* mutants.

## Results

### Successful segmentation of each somite is randomly determined in clock mutants

To quantify intact and defected segmentation, we performed in situ hybridization (ISH) for *xirp2a*, a marker for somite boundaries[19]. Similar to a previously published *her1^hu2124* line[15], we found that sibling *her1^ci302* mutants displayed incomplete penetrance (Fig. 1b): 15% of mutants successfully segmented all somites whereas 85% of them displayed defected boundaries (Fig. 1c, d, median defected boundary = 1). On the other hand, *her7^hu2526* mutants displayed complete penetrance of segmentation defects (Fig. 1e), confirming earlier results[15]. However, we additionally found that *her7^hu2526* embryos displayed variable expressivity (i.e., phenotypic strength among siblings); not all boundaries were broken in the posterior domain as intact and broken boundaries were intermingled (Fig. 1e). Because these embryos are siblings and were raised under the same environment, we hypothesized that the fundamental source of their phenotypic variation is likely stochastic gene expression[20]. If the remaining functional clock gene (i.e., *her7* in *her1^ci302* or *her1* in *her7^hu2526* mutants) stochastically establishes a limit cycle in groups of cells but fails to do so in others, then it would result in randomly distributed intact and defected somites along the axis. To test this hypothesis, we scored the phenotype of each boundary (as defected = 1 and intact = 0) and calculated the cross-correlation of phenotypes of consecutive somites starting from the 11th boundary until the 30th one on each side (left or right) of individual *her7^hu2526* mutants (Fig. 1f). The average cross-correlation score was close to zero (0.05, Fig. 1g). This result suggests that the phenotypic outcomes of consecutive somites on each side of an

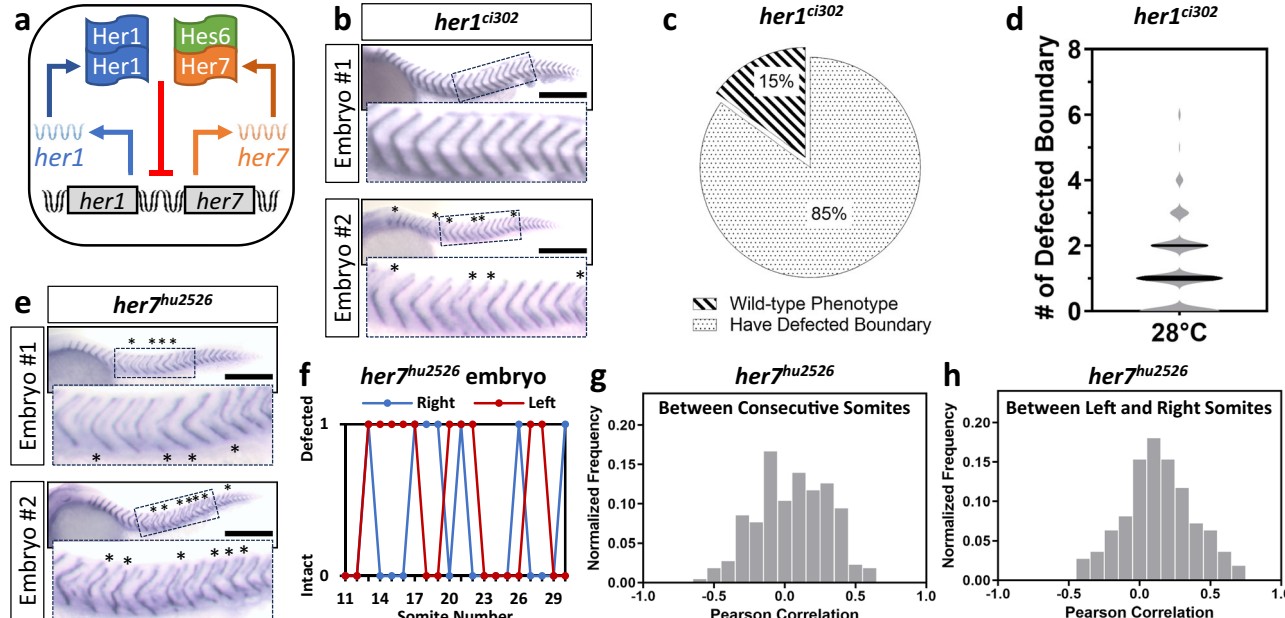

**Fig. 1 | Segmentation defects in mutants are stochastic. a** Her1-Her1 and Her7-Hes6 dimers form a negative feedback loop by repressing their own transcription. **b** Somite boundaries are marked by *xirp2* ISH staining in sibling *her1^ci302* mutant embryos. While the upper embryo has no segmentation defect, lower embryo has 6 defective segmentation boundaries (marked by the star). Insets show 500-µm-wide enlarged images of boundaries. Scale bar is 300 µm. **c, d** Percentage of *her1^ci302* mutants displaying wild-type phenotype (striped) or segmentation defects (dotted) at 28 °C (**c**, *n* = 50, *N* = 15), and number of defective boundaries per each side of *her1^ci302* mutants (**d**, *n* = 100, *N* = 15). The violin plot shows the median (thick black line) and quartiles (thin black lines). **e** *xirp2* ISH staining of *her7^hu2526* mutants. Insets show 500-µm-wide enlarged images of boundaries. Scale bar is 300 µm. **f** Scoring of segmentation defect position of one *her7^hu2526* mutant embryo raised at 28 °C. **g** The Pearson correlation of same side consecutive somite phenotype (*n* = 222, *N* = 3), and (**h**) same position left- and right-side somite phenotype (*n* = 111, *N* = 3) between 11th and 30th somite boundaries. *n* is the number of embryos in (**c, h**), and sides in (**d, g**); *N* is the number of fish pairs in (**c, d**), and number of independent experiments in (**g, h**). Source data are provided as a Source Data file.

embryo are independently determined (e.g., like flipping a coin). In wild-type embryos, somite segmentation is bilaterally symmetric[10]. In contrast, the average Pearson correlation of phenotypic score was close to zero (0.13) for left and right somites at the same position in *her7*[hu2526] mutants (Fig. 1h). This result shows that somite segmentation on the left and right sides become approximately independent of each other in *her7*[hu2526] mutants (Fig. 1f). Because cells segmenting into different somites are genetically identical, these results strongly suggest that variable segmentation defects are stochastically set.

### Transcriptional correlation of clock genes decreased in segmentation clock mutants

We next sought to quantify the correlation of *her1* and *her7* transcription in three genetic backgrounds (wild-type, *her1*[ci302] and *her7*[hu2526] mutants). We performed single molecule fluorescence in situ hybridization (smFISH) to simultaneously count the *her1* and *her7* RNAs (Fig. 2a–c) of each cell in intact PSM tissues[13,21]. The period of oscillations of the segmentation clock is the fastest in cells located in the posterior PSM[22,23]. As cells are displaced to more anterior locations, the period of the clock gradually slows down. Due to these

spatiotemporal dynamics, clock RNA and protein levels display posterior to anterior kinematic waves along the PSM. Therefore, at any time snapshot (i.e., fixed smFISH sample), two to three stripes of RNA transcription are detectable along the PSM during early somitogenesis stages. Normally, in wild-type embryos, *her1* and *her7* RNA stripes are mostly in phase in the anterior PSM (Fig. 2a). In contrast, clock expression stripes became more variable in *her1*[ci302] and *her7*[hu2526] mutants (Fig. 2b, c).

The spatial Pearson correlation of *her1* and *her7* is a useful metric to measure the oscillation quality in each genotype[13]. In this manner, we assessed the correlation of spatial coexpression of *her1* and *her7* RNAs among different genotypes using single transcript counts obtained by smFISH data. We found that the correlation score significantly decreased in both *her1*[ci302] and *her7*[hu2526] mutants as compared to wild-type embryos (Fig. 2d; Supplementary Fig. 1). But the Pearson correlation score could not differentiate between *her1*[ci302] and *her7*[hu2526] mutants; this could either be due to this metric reaching its resolution limit in *her1*[ci302] mutant or the difficulty of precisely measuring the expression stripe angles in the clock mutants (Supplementary Fig. 2).

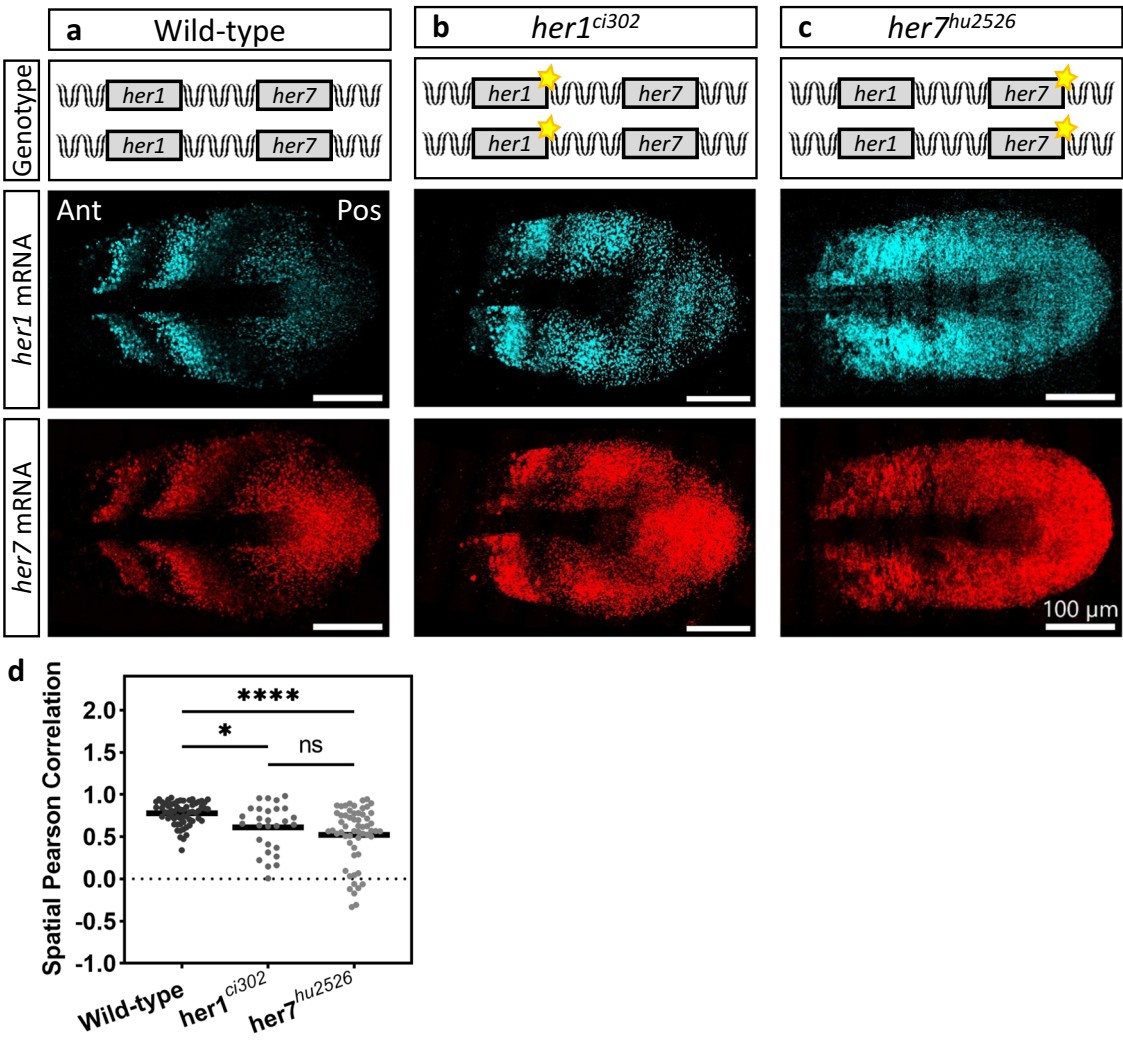

**Fig. 2 | Gene expression in mutants are stochastic. a–c** smFISH was performed at 12–14 somite stage to count *her1* (cyan) and *her7* (red) transcription in wild-type (**a**), *her1*[ci302] (**b**), and *her7*[hu2526] (**c**) embryos raised at 23 °C. Scale bar is 100 μm. Yellow stars mark *her1* and *her7* genes carrying point mutations causing premature stop codons. **d** The spatial Pearson correlation scores of *her1* and *her7* in wild-type (dark gray, $n = 31$, $N = 3$), *her1*[ci302] (gray, $n = 14$, $N = 2$), and *her7*[hu2526] (light gray, $n = 29$,

$N = 2$) embryos. *$P = 0.0157$ (wild-type versus *her1*[ci302]), ****$P = 0.2146 \times 10^{-5}$ (wild-type versus *her7*[hu2526]), $P = 0.7243$ (*her1*[ci302] versus *her7*[hu2526]), Kruskal–Wallis ANOVA with Dunn's multiple-comparison correction. ns, not significant. Black lines show the mean of data in the scatter dot plot. $n$ is the number of embryos; $N$ is the number of independent experiments. Source data are provided as a Source Data file.

## Successful segmentation is preceded by high amplitude clock oscillations

We then hypothesize that an amplitude threshold of oscillations is required for successful segmentation. In certain genetic backgrounds, this threshold may not stochastically be achieved during some clock cycles, resulting in randomly distributed segmentation defects intermingled with successful segmentation in each embryo. To test the threshold hypothesis, we sought to quantify the amplitude of oscillations by imaging clock reporters in real-time in a genetic background displaying variable expressivity and containing only the reporters as the functional clock genes. *her1-Venus*[24] and *her7-Venus*[25] reporter transgenes are inserted at different chromosomes from the endogenous locus. Due to loss of both clock genes, *her1ci301;her7hu2S26* mutants fail to properly segment all segment boundaries[13]. We found that double mutant embryos carrying both transgenes at single chromosomes (*her1ci301;her7hu2S26;her1-Venus+/-;her7-Venus+/-*) rescued segmentation of some but not all somite boundaries and thereby generated an excellent model displaying variable expressivity (Fig. 3a–c). Between 11th and 25th somites, the median number of defected boundaries was 8 (Fig. 3d). Because endogenous clock proteins are absent in this genetic background, successful segmentation

must be due to the reporter transgenes. We performed an smFISH experiment by using a probe against *Venus* and counted the total number of *her1-Venus* and *her7-Venus* RNAs in this genetic background (Supplementary Fig. 3a). We found that the mean of transgene RNAs in mutant embryos was 36% lower than the mean of endogenous (*her1* plus *her7*) RNAs in wild-type embryos (Supplementary Fig. 3b), suggesting the reporter transgenes are at least partially functional.

We imaged these embryos for 10 h at 26 °C (Fig. 3e), and generated kymographs of the fluorescence signal coming from clock reporters (Fig. 3f). Segmental commitment occurs in the middle of the PSM at a position called the determination front[10], where the clock instructs segment boundaries[22,25]. Loss of clock waves or expression in the aPSM could not stop segmentation[22] and the kinematic clock waves are irrelevant for somite boundary formation[25]. We previously identified the position of the determination front throughout somitogenesis[26]. We therefore quantified the total reporter signal along the trajectory of determination front (Fig. 3g, blue line along kymograph; Supplementary Fig. 4). Afterwards, we calculated the amplitude of oscillations at the determination front. In parallel, we scored which segment boundaries are successfully formed or not. We then separately grouped amplitudes corresponding to intact or

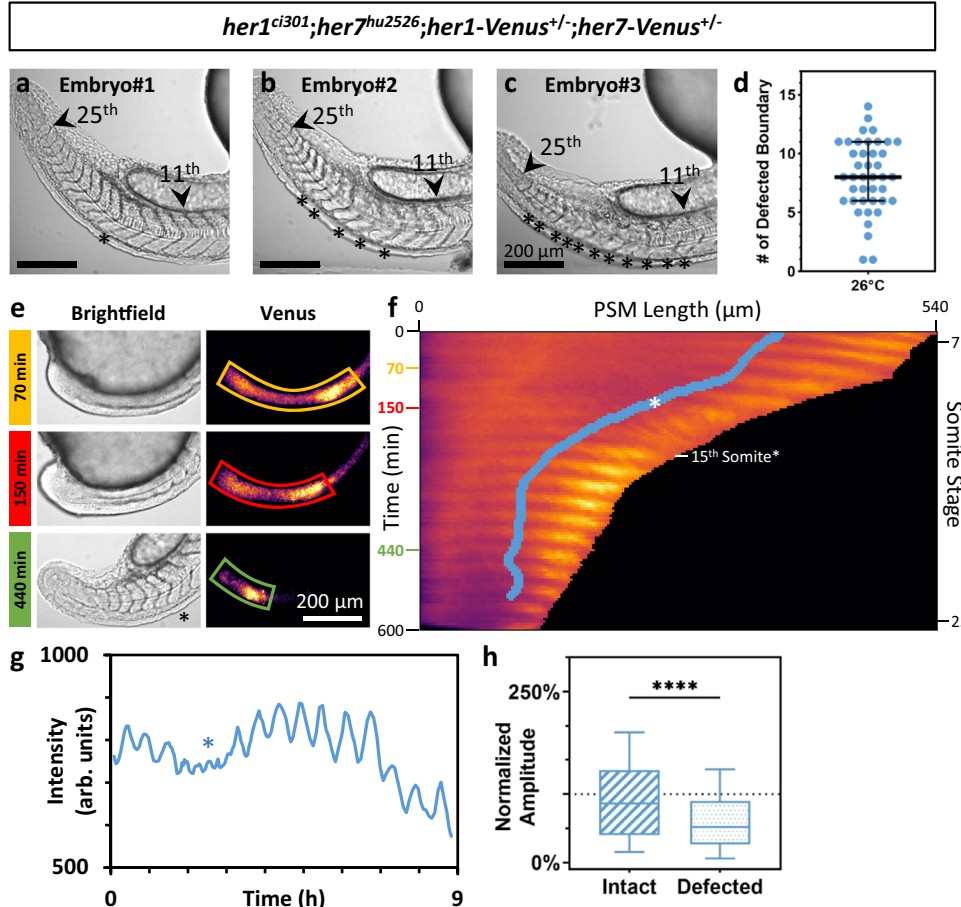

**Fig. 3 | An amplitude threshold of oscillations is required for successful segmentation in a genetic background displaying variable expressivity.**
**a–c** Brightfield images of *her1ci301;her7hu2S26;her1-Venus+/-;her7-Venus+/-* embryos with variable expressivity. Representative embryos are ordered by their increasing number of defective somites (**a–c**). Stars show defected boundaries between 11th and 25th somite boundaries. **d** Number of defected boundaries per one side of each embryo (*n* = 41, *N* = 3). Black lines show median (thick), and quartiles (thin).
**e** Brightfield and Venus images of a single embryo at 70, 150, and 440 min. Yellow, red and green lines show line-of-interest (LOI) borders in Venus channel at each time point. Star shows a defected boundary. **f** A representative kymograph of Venus signal seen in (**e**) between 7 and 25 somite stages (for 600 min, *y*-axis) along the

entire PSM (*x*-axis, posterior-aligned to the left). Blue solid line shows determination front position. Star shows where and when the 15th boundary (defected) is determined, and white dash shows when it is formed. **g** Smoothened line profile of Venus signal along determination front position (arbitrary units, arb. units). Blue star shows the wave determining the defected boundary. **h** Normalized amplitude of waves for intact (striped, *n* = 283, *N* = 3) and defected (dotted, *n* = 328, *N* = 3) boundaries. ****$P = 0.4908 \times 10^{-8}$, two-tailed Mann–Whitney *U*-test. The whisker plot shows the median (line), quartiles (box), as well as the 10th and 90th percentiles (whiskers). *n* is the number of embryos in (**d**), and boundaries in (**h**); *N* is the number of independent experiments. Scale bars are 200 μm. Source data are provided as a Source Data file.

defected segmentation. We found that the median amplitude preceding defected boundaries is 40% less than those preceding intact ones (Fig. 3h; Supplementary Fig. 5). This result shows that higher oscillation amplitude correlates with successful somite segmentation.

The decrease in average amplitude of oscillations could be due to decreases of oscillation amplitudes in individual cells or desynchronization of oscillations among cells as seen when Delta/Notch signaling is weakened in the PSM[19,27]. We do not favor the latter scenario because Her1 and Her7 repress transcription of *deltaC* ligand[22] and therefore *her1* and *her7* clock mutants are expected to increase Delta/Notch signaling. Nonetheless, similar phenotypic variability between the left and right sides of embryos can also occur when Notch signaling is transiently pharmaceutically impaired[28].

### Environmental factors can modify the strength of segmentation phenotypes

Environmental factors are associated with congenital vertebral malformations in patients[2–4]. We have previously shown that transcriptions of *her1* and *her7* are more correlated and less noisy at higher, rather than lower, temperatures in wild-type embryos[13]. Furthermore, the severity of the segmentation defects increased in a different (*hes6*) mutant at low temperatures[29,30]. Therefore, we next investigated the impact of temperature on the variable phenotypes in *her1*[ci302] or *her7*[hu2S26] mutants. We found that decreasing temperature from 28 °C to 23 °C increases penetrance of *her1*[ci302] mutant segmentation defect from 85% to 99%. We found that a further decrease in growing temperature to 21 °C switches the phenotype of *her1*[ci302] mutant from incomplete penetrance to fully penetrant but with variable expressivity (Fig. 4a). In addition, the median number of defected boundaries increased from 1 at 28 °C to 3 at 23 °C, and then to 5 at 21 °C in *her1*[ci302] mutants (Fig. 4b). When *her7*[hu2S26] mutants were raised at various temperatures during somitogenesis, we consistently saw gradually reduced expressivity of segmentation defects at higher temperatures (Fig. 4c). The median number of defected boundaries increased from 8 at 28 °C to 10 at 23 °C, and then to 13 at 21.5 °C, an overall 63% increase compared to the highest temperature (Fig. 4c). These results show that the phenotypic strength of *her1*[ci302] and *her7*[hu2S26] mutants can be quantitatively changed by temperature, whereas no

segmentation defects were seen in wild-type embryos raised at low temperatures (Fig. 4a).

Exposure to hypoxia during pregnancy induces congenital vertebral malformations in humans[3,4]. Likewise, gestational hypoxia causes vertebral segmentation defects in mutants of Notch signaling pathway genes (e.g., *Hes7*[+/-]) in mice[2]. Therefore, we next treated *her7*[hu2S26] embryos with $CoCl_2$, which was previously shown to induce hypoxia in zebrafish[31]. We found that $CoCl_2$ treatment increased the average number of segmentation defects by 29% (8.9 vs 11.5) in *her7*[hu2S26] mutants raised at 28 °C (Fig. 4d). Overall, these data showed that environmental perturbations, such as low temperature and hypoxia, can augment the phenotypic strength observed in clock mutants.

We next quantified the amplitude of oscillations in the cells located at the determination front and the segmentation phenotype of those corresponding cells in *her1*[ci301];*her7*[hu2S26];*her1-Venus*[+/-];*her7-Venus*[+/-] embryos grown at a lower temperature of 21.5 °C or grown at 26 °C but under hypoxic conditions (Fig. 5a–f). We found that in both unfavorable conditions, the amplitude of oscillations preceding successful segmentation were higher than those preceding defective segmentation (Fig. 5g, h). Importantly, the number of cycles with high amplitudes and the number of successful segmentations are proportionally decreased in embryos grown in unfavorable environments as compared to 26 °C controls (Fig. 5i, j).

Besides natural environmental factors, like temperature and hypoxia, other embryotoxic chemicals were previously shown to worsen segmentation defects in zebrafish[32]. We confirmed that two of those chemicals (IPA-3 and SB225002) cause segmentation defects at moderate (3 μM) (Fig. 6a, b) but not low (1 μM) (Supplementary Fig. 6) doses in wild-type embryos. Since previous work showed that these two chemicals do not directly affect the segmentation clock[32], these drugs might impair processes occurring after segmental commitment. Future studies may investigate the specific molecular pathways that are affected by these chemicals.

### Discussion

While gene expression variability is confined within a tolerable range in wild-type embryos, it can be untamed in mutant conditions. Although homozygous mutations in the segmentation network genes result in

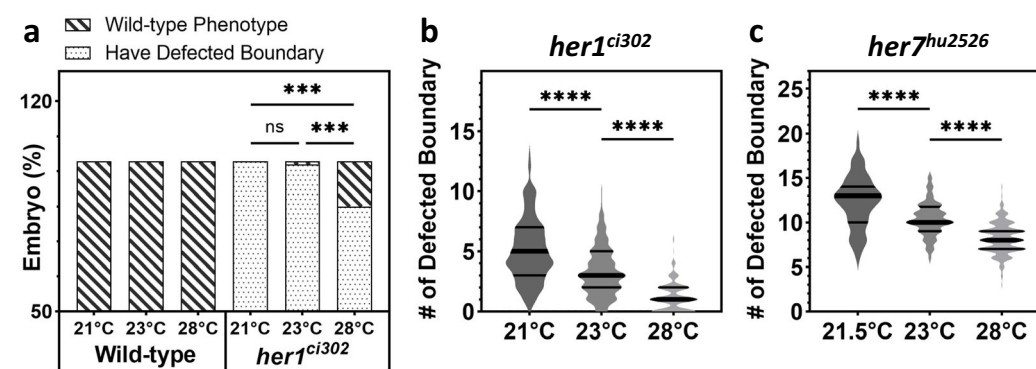

**Fig. 4 | Environmental factors affect the strength of segmentation phenotypes.**
**a** Embryo percentage of wild-type and *her1*[ci302] mutants raised at 21 °C, 23 °C, and 28 °C based on segmentation phenotype. Wild-type embryos have intact boundaries at all temperatures ($n = 100$, $N = 12$). *her1*[ci302] mutants displaying segmentation defects at 21 °C ($n = 78$, $N = 15$), but incomplete penetrance phenotypes at 23 °C ($n = 99$, $N = 15$), and 28 °C ($n = 100$, $N = 15$). Striped bar denotes embryos with wild-type phenotype, dotted bar denotes defected embryos. $P > 0.9999$, \*\*\*$P = 0.0002$ and \*\*\*$P = 0.0003$ between embryos raised at 21 °C and 23 °C, 21 °C and 28 °C, and 23 °C and 28 °C, respectively (two-tailed Fisher's exact test). ns, not significant.
**b** Number of defective boundaries per each side of *her1*[ci302] mutants at different temperatures, 21 °C ($n = 156$, $N = 15$), 23 °C ($n = 198$, $N = 15$), and 28 °C ($n = 200$, $N = 15$). \*\*\*\*$P = 0.5355 \times 10^{-7}$ (21 °C vs 23 °C), \*\*\*\*$P < 0.0001 \times 10^{-11}$ (23 °C vs 28 °C),

Kruskal–Wallis ANOVA with Dunn's multiple-comparison correction. **c** Number of defective boundaries per each side of *her7*[hu2S26] mutants at 21.5 °C ($n = 58$, $N = 1$), at 23 °C ($n = 192$, $N = 4$), and at 28 °C ($n = 198$, $N = 4$). \*\*\*\*$P = 0.1305 \times 10^{-4}$ (21.5 °C vs 23 °C), \*\*\*\*$P < 0.0001 \times 10^{-11}$ (23 °C vs 28 °C), unpaired two-tailed ANOVA without equal standard deviation assumption and with Games-Howell's multiple comparison test. **d** Hypoxia ($n = 300$, $N = 4$) affects somite segmentation phenotype (controls: $n = 300$, $N = 4$). \*\*\*\*$P < 0.0001 \times 10^{-11}$, two-tailed Mann–Whitney $U$-test. The violin plots show the median (thick black line) and quartiles (thin black lines). $n$ is the number of embryos in (**a**), and sides in (**b**–**d**); $N$ is the number of fish pairs in (**a**–**c**), and number of independent experiments in (**d**). Source data are provided as a Source Data file.

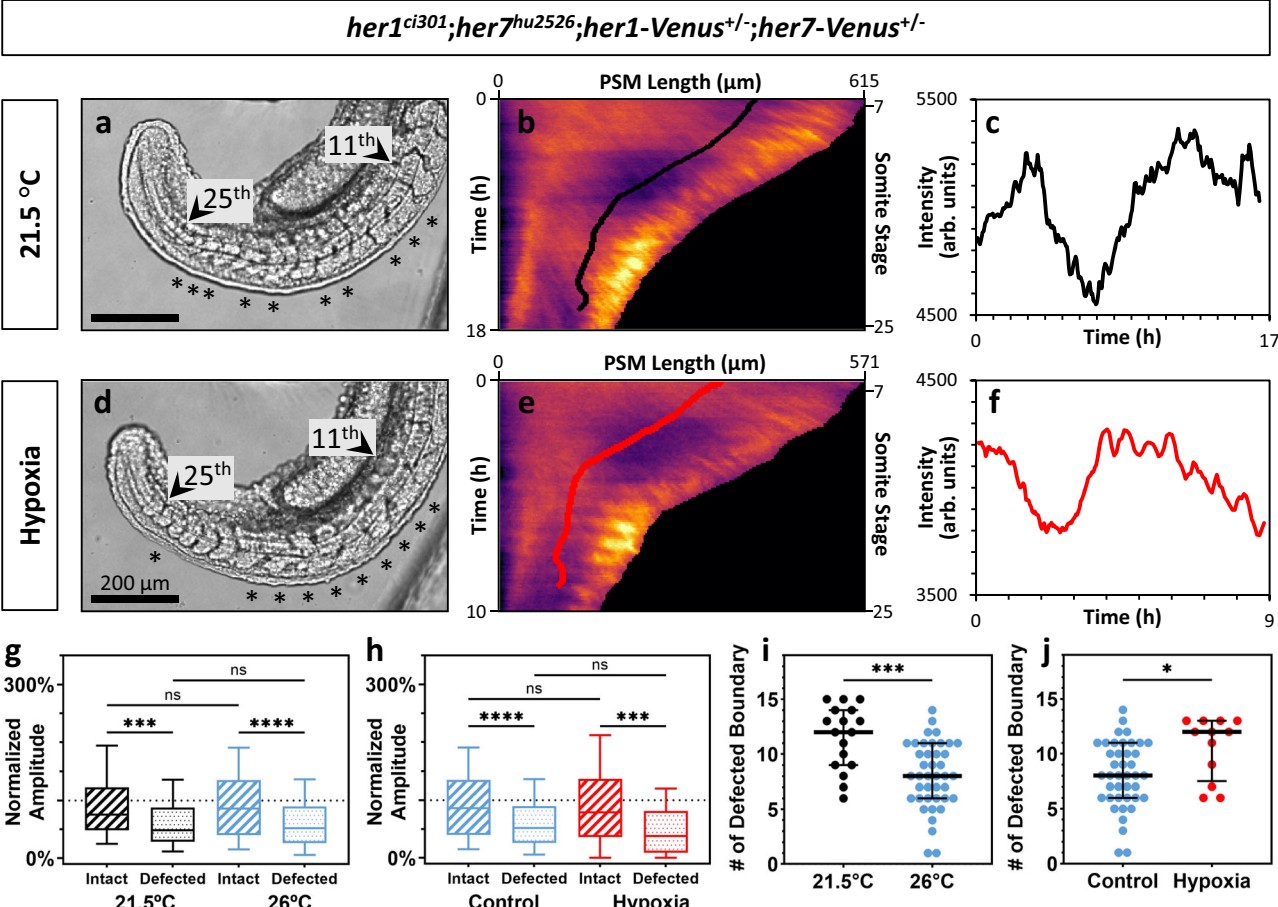

**Fig. 5 | Segmentation clock amplitudes correlate with segmentation phenotypes affected by environmental factors. a–f** Brightfield images (**a–d**) kymographs (**b–e**), and line profiles (**c–f**) along the positions of determination fronts seen on the kymographs for representative *her1^ci301^;her7^hu2526^;her1-Venus^+/-^;her7-Venus^+/-^* embryos raised at 21.5 °C (**a–c**, black line), and in hypoxia environment at 26 °C (**d–f**, red line) (arbitrary units, arb. units). **g, h** Normalized amplitudes of waves for intact (black striped, *n* = 59, *N* = 2) and defected (black dotted, *n* = 196, *N* = 2) boundaries, and intact (red striped, *n* = 53, *N* = 2) and defected (red dotted, *n* = 127, *N* = 2) boundaries of embryos raised at 21.5 °C (**g**), and in hypoxia environment at 26 °C (**h**), respectively, compared with the control embryos (blue) raised at 26 °C. ****P = 0.1496 x 10⁻⁷ (26 °C intact vs 26 °C defected), ***P = 0.0008 (21.5 °C intact vs 21.5 °C defected), P > 0.9999 x 10⁻¹¹ (21.5 °C intact vs 26 °C intact), P > 0.9999 x 10⁻¹¹

(21.5 °C defected vs 26 °C defected), ****P = 0.7192 x 10⁻⁷ (control intact vs control defected), ***P = 0.0007 (hypoxia intact vs hypoxia defected), P = 0.0678 (control defected vs hypoxia defected), P > 0.9999 x 10⁻¹¹ (control intact vs hypoxia intact), Kruskal–Wallis ANOVA with Dunn's multiple-comparison correction. ns, not significant. **i, j** Number of defected boundaries per one side of each embryo raised at 21.5 °C (*n* = 17, *N* = 2) (**i**) and in hypoxic (*n* = 12, *N* = 2) conditions (**j**). Black lines show median (thick), and quartiles (thin). ***P = 0.0002 (21.5 °C vs 26 °C), *P = 0.0135 (control vs hypoxia) two-tailed unpaired *t*-test. The whisker plots show the median (line), quartiles (box), as well as the 10th and 90th percentiles (whiskers). *n* is the number of boundaries in (**g, h**), and embryos in (**i, j**); *N* is the number of independent experiments. Source data are provided as a Source Data file.

---

penetrant scoliosis, heterozygosity results in phenotypic variability[2]. Usually, human geneticists and clinicians generally attribute variable outcomes of a phenotype to differences in genetic backgrounds (modifier gene concept) or gene-environment interactions[2–6]. Although these factors could amplify phenotypic variability, our analysis within single embryos of zebrafish segmentation clock mutants highlighted the importance of stochastic gene expression: the randomness of segmentation decision among populations of genetically identical cells in each embryo strongly suggests that the fundamental source of phenotypic variability (both incomplete penetrance and variable expressivity) is stochastic gene expression.

We also showed here that environmental factors could modify the strength of phenotype. Hypoxia increased the strength of segmentation defects in *her7^hu2526^* mutants. Moreover, we could increase the expressivity of segmentation phenotype in *her7^hu2526^* mutant by decreasing the temperature, and vice versa. Similarly, by decreasing the incubation temperature, we were able to increase the penetrance of the

segmentation phenotype in *her1^ci302^* mutants such that the phenotype became completely penetrant. These results suggest that variable expressivity and incomplete penetrance are two sides of the same coin, which can be flipped under specific environmental conditions.

We additionally showed that the amplitude of protein oscillations is higher in groups of cells successfully committing to segmentation at the determination front than those failing to do so. Because Her1 and Her7 proteins repress their own and each other's transcription, their expression is expected to be derepressed in the mutant of each gene. However, derepression of the functional clock genes (e.g., *her7* in *her1^ci302^* or *her1* in *her7^hu2526^* mutants) will stochastically differ across the PSM cells. In single *her1^ci302^* or *her7^hu2526^* mutants, the clock function stochastically fails during some cycles, resulting in low collective amplitudes among a population of cells. This failure then results in a less robust pattern and variable somite boundary formation as compared to wild-type embryos. These results suggest that genetic redundancy partially counteracts expression variability, preventing variable phenotypes.

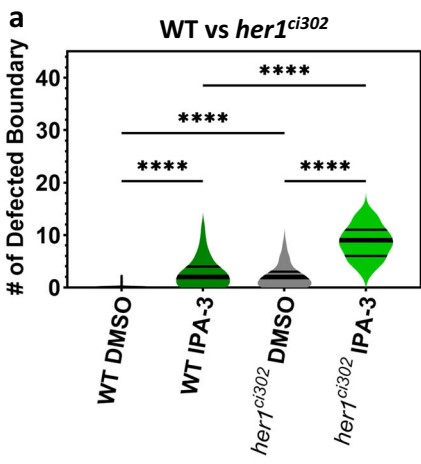

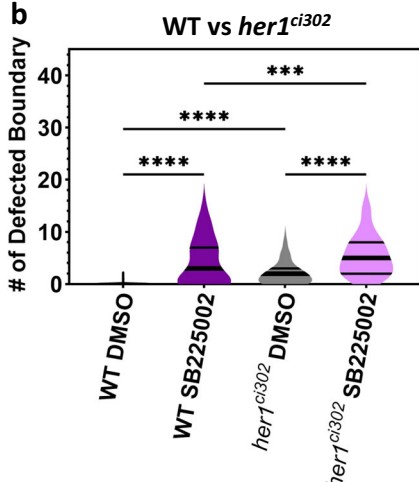

**Fig. 6 | Embryotoxic chemicals worsen segmentation defects. a, b** Number of defective boundaries per each side of wild-type (WT) and *her1*<sup>ci302</sup> mutants treated with 3 μM DMSO (**a, b**, gray) (*n* = 172, *N* = 3, and *n* = 156, *N* = 3), 3 μM IPA-3 (**a**, green) (*n* = 138, *N* = 3, and *n* = 172, *N* = 3), and 3 μM SB225002 (**b**, purple) (*n* = 110, *N* = 3, and *n* = 168, *N* = 3). ****P* < 0.0001 × 10⁻¹¹ (WT DMSO vs WT IPA-3), ****P* < 0.0001 × 10⁻¹¹ (WT DMSO vs *her1*<sup>ci302</sup> DMSO), ****P* < 0.0001 × 10⁻¹¹ (WT IPA-3 vs *her1*<sup>ci302</sup> IPA-3), ****P* < 0.0001 × 10⁻¹¹ (*her1*<sup>ci302</sup> DMSO vs *her1*<sup>ci302</sup> IPA-3), ****P* < 0.0001 × 10⁻¹¹ (WT

DMSO vs WT SB225002), ****P* < 0.0001 × 10⁻¹¹ (WT DMSO vs *her1*<sup>ci302</sup> DMSO), ***P* = 0.0001 (WT SB225002 vs *her1*<sup>ci302</sup> SB225002), ****P* = 0.4647 × 10⁻¹¹ (*her1*<sup>ci302</sup> DMSO vs *her1*<sup>ci302</sup> SB225002), Kruskal–Wallis ANOVA with Dunn's multiple-comparison correction. The violin plots show the median (thick black line) and quartiles (thin black lines). *n* is the number of sides; *N* is the number of independent experiments. Source data are provided as a Source Data file.

Understanding the sources of incomplete penetrance and variable expressivity of phenotypes will enhance both our knowledge of developmental genetics and the progress toward personalized medicine. If we can understand why phenotypes only sometimes manifest in mutants, we may someday capitalize on natural mechanisms of genetic resilience for human therapies.

## Methods
### Animals
The zebrafish experiments were performed under the ethical guideline of Cincinnati Children's Hospital Medical Center. The animal protocol was reviewed and approved by Cincinnati Children's Hospital Medical Center Animal Care and Use Committees (Protocol # 2020-0031). In this study, we used AB wild-type line, *her7*<sup>hu2S26</sup> [18], *her1*<sup>ci302</sup> [13], *her1*<sup>ci301</sup>;*her7*<sup>hu2S26</sup> [13] mutant lines, and Tg(*her1:her1-Venus*)[24] and Tg(*her7:her7-Venus*)[25] transgenic lines. Fish were bred and maintained at 28.5 °C on a 14/10-hour light/dark cycle. After fertilization, embryos were kept in E3 medium. To assess temperature effects, embryos were incubated at 28 °C until shield stage, then transferred to a lower temperature. Live images were done at 21.5 °C or 26 °C. Drug treated embryos were kept at 28 °C until fixation. Embryos were manually dechorionated using needle tips.

### In situ hybridization
We performed *xirp2a* (cb1045) probe in situ staining to label somite boundaries[19]. Digoxigenin-labeled RNA probes were prepared by in vitro transcription and anti-digoxigenin (DIG)-AP Fab fragments (Roche, 1093274) were used for detection. Embryos were kept at 28 °C until shield stage, and then transferred to a temperature range between 21 °C and 28 °C. To prevent pigmentation, embryos were treated with 0.003% *N*-phenylthiourea (PTU) (Sigma, P7629) after the end of somitogenesis. They were later fixed at 40 hpf developmental stage with 4% PFA in PBS at room temperature for 2 h. NBT/BCIP (Roche, 1168145100) stained samples were imaged with a Nikon SMZ1500 stereomicroscope (HR Plan Apo 1X WD 54) equipped with SPOT Imaging Insight CMOS camera and reflected light at room temperature. Boundaries up to 30 were scored as intact if they are completed or as defected if they are fragmented or incomplete. If an embryo has a single defected boundary on either side, it is scored as defected.

### Pharmacological treatments
Cobalt chloride (CoCl₂) (Sigma, C8661) was used in hypoxia experiments. It was dissolved in embryo medium at a concentration of 192 mg/mL. *her7*<sup>hu2S26</sup> embryos raised at 28 °C were treated with 10 mM cobalt chloride (CoCl₂) at shield stage and kept at 28 °C until 40 hpf before fixation. *her1*<sup>ci301</sup>;*her7*<sup>hu2S26</sup>;*her1-Venus*<sup>+/-</sup>;*her7-Venus*<sup>+/-</sup> embryos were incubated at 26 °C with 10 mM CoCl₂ from shield stage to the end of live imaging. IPA-3 (Sigma, I2285) and SB225002 (Sigma, SML0716) were dissolved in DMSO at 10 mM. Wild-type and *her1*<sup>ci302</sup> embryos were raised at 28 °C and were treated with 1 μM and 3 μM IPA-3 and SB225002 at bud stage for embryotoxic environmental conditions. 3 μM DMSO was used as a control.

### smFISH experiments and imaging
We performed smFISH experiments and confocal imaging following our optimized protocol[33,34]. Membrane-localized GFP was injected into *her1*<sup>ci302</sup> and *her7*<sup>hu2S26</sup> mutant embryos, and membrane-localized mCherry was injected into *her1*<sup>ci302</sup>;*her7*<sup>hu2S26</sup>;*her1Venus*<sup>+/-</sup>;*her7Venus*<sup>+/-</sup> transgene embryos at single cell stage. Dr-*her1*-LE2-C3 and Dr-*her7*-C1 were used to detect *her1* and *her7* mRNAs in *her1*<sup>ci302</sup> and *her7*<sup>hu2S26</sup> mutant embryos; and Dr-EGFP-C3 probes were used to detect both *her1Venus* and *her7Venus* (total *Venus*) mRNAs in transgene embryos. Embryos were incubated at 23 °C and fixed at 12–14 somite stage. Chicken monoclonal IgY anti-GFP (Cat# A10262, Invitrogen, Lot# 2321831, 1:200) and rabbit polyclonal Living Colors anti-DsRed (mCherry; Cat# 632496, Takara, Lot# 2103116, 1:200) were used as primary antibodies. Alexa Fluor 488 goat anti-chicken IgG H + L (Cat# A11039, Invitrogen, Lot# 2420700, 1:200) and Alexa Fluor 594 donkey anti-rabbit IgG H + L (Cat# A21207, Invitrogen, Lot# 2066086, 1:200) were used as secondary antibodies. We imaged 7 additional wild-type embryos in addition to 24 wild-type samples previously reported in[13], and analyzed 31 wild-type embryos in total. We imaged and analyzed 14 *her1*<sup>ci302</sup>, 29 *her7*<sup>hu2S26</sup>, and 18 *her1*<sup>ci301</sup>;*her7*<sup>hu2S26</sup>;*her1-Venus*<sup>+/-</sup>;*her7-Venus*<sup>+/-</sup> embryos. The total number of analyzed cells and slices were 63,142 and 3765 in wild-type, 22,706 and 1717 in *her1*<sup>ci302</sup>, 45,791 and 3260 in *her7*<sup>hu2S26</sup> mutants, and 24,010 and 1945 in *her1*<sup>ci301</sup>;*her7*<sup>hu2S26</sup>;*her1-Venus*<sup>+/-</sup>;*her7-Venus*<sup>+/-</sup> transgene, respectively. Background subtracted RNA levels were used to calculate the means of RNA transcripts. Imaris 9.8 was used for cell segmentation analysis.

## Calculating sample cross-correlation and spatial Pearson correlation coefficients

Pearson product-moment correlation coefficient was used to compare phenotypes of consecutive somite boundaries between 11th–29th and 12th–30th of each side, and to compare phenotypic scores between 11th and 30th boundaries of left- and right-sides in *her7*[hu2526] mutants. Pearson correlation scores were later binned to 0.1 and plotted as normalized frequency.

We also quantified transcriptional covariance of *her1* and *her7* in space of wild-type, and *her1*[ci302] and *her7*[hu2526] mutants by Pearson correlations coefficients. As ~40–80% of PSM corresponds to regions of characteristic kinematic clock waves, we used the slice averages of *her1* and *her7* mRNA concentrations along 40–80% of PSM from the posterior end[13]. We measured the angles of expression stripes in wild-type and *her1*[ci302] mutants, fitted an equation to the data, and used them to divide tissue into separate slices[13]. Measuring the stripe angles throughout the PSM in *her7*[hu2526] mutants and *her1*[ci301];*her7*[hu2526];*her1-Venus*[+/-];*her7-Venus*[+/-] transgene is difficult. Therefore, we used the mean of the angles measured at the anterior-most expression domain as a fixed angle for slicing the PSM in these mutants.

## Time-lapse imaging and image analysis

We imaged *her1*[ci301];*her7*[hu2526];*her1-Venus*[+/-];*her7-Venus*[+/-] embryos at 21.5 °C or 26 °C. Up to 14 dechorionated embryos per experiment were imaged laterally in holes in 1% agarose gel (in E3 medium with 4% tricaine) using 15-mm glass-bottom dish (Azer Scientific, ES56291). Images were captured under Nikon Ti-E SpectraX Widefield Microscope (Plan Apo λ 10X) at a 512×512-pixel resolution (pixel size: 2.60 μm). The samples were excited by 508-nm laser and images were recorded in brightfield and YFP (535/30 filter) channels with 5 z-slices in 30-μm steps and a time interval of 5 min for at least 10 h. Posterior (tail bud) aligned kymographs were generated from the YFP images between 7 and 25 somite stages using the LOI interpolator (line width set to 15-px, 39 μm). The positions of the determination fronts were determined on kymographs using the posterior PSM to full PSM ratio as reported in[26]. After plotting 4-px-width of line profile at these positions in each kymograph, the raw intensity (arbitrary units, arb. units) was smoothened with Savitzky-Golay filter and resampled 10 times using Matlab. Amplitudes of 15 consecutive waves were calculated as the intensity difference between trough to peak starting from 7 somite stage. Peaks and troughs for each wave were detected from local maxima and minima, respectively, within a time window of approximate clock period (i.e., 12 frames for 21.5 °C and seven frames for 26 °C). If no trough or peak is detected in a periodic window, amplitude is set to zero. The first clock wave passing at 7 somite stage determines the 11th somite boundary[26]. Boundary phenotypes are scored by going over the z-sections of brightfield images. Posterior boundary phenotypes of the somites (starting from the 11th) were paired with the corresponding amplitudes specifying those boundaries. Then, each amplitude is normalized to the mean amplitude of intact boundaries of each independent experiment. Nikon NIS Elements 9.0, FIJI (ImageJ 1.54 f), and Matlab 2022b were used for the image analysis, and GraphPad Prism 9.5.0 and Microsoft 365 Excel were used to plot graphs.

## Statistical analysis

We used parametric ANOVA (two-tailed, without equal standard error assumptions with Welch and Brown-Forsythe's method and Games-Howell multiple comparison test) in Fig. 4c, and two-tailed unpaired *t*-test in Fig. 5i, j as distributions passed the D'Agostino-Pearson normality test. As for the non-parametric distributions, we ran Mann–Whitney *U*-test to determine whether there were differences between two groups in Figs. 3h, 4d, Supplementary Fig. 5d–f, and we ran Kruskal–Wallis ANOVA nonparametric tests with Dunn's multiple correction test in Figs. 2d, 4b, 5g, h, 6a, b, and Supplementary Fig. 2a–c,

6a, b. To compare *her1*[ci302] embryos with or without any defected boundary raised at various temperatures (Fig. 4a), we ran two-tailed Fisher's exact test between every two conditions. All the statistical tests and distribution calculations (median and quartiles, confidence intervals) were performed in GraphPad Prism 9.5.0 software.

## Reporting summary

Further information on research design is available in the Nature Portfolio Reporting Summary linked to this article.

## Data availability

The microscopy images generated in this study have been deposited in the BioStudies database under accession code S-BSST1156. There are no restrictions on data availability. Source data are provided with this paper.

## Code availability

Custom Matlab and FIJI codes are available at GitHub (https://github.com/ozbudak/Keseroglu_2023_Penetrance), which is archived in Zenodo with the identifier (https://doi.org/10.5281/zenodo.8377968)[35].

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

## Acknowledgements
We thank M. Batie, M. Kofron, the staff at Cincinnati Children's Imaging Core and Cincinnati Children's Veterinary Services for technical assis- tance; Hilal Cevik and Amanda Zacharias for providing feedback on the manuscript. This work was funded by a US National Institute of General Medical Sciences of the National Institutes of Health grant R35GM140805 to E.M.Ö.

## Author contributions
E.M.Ö. conceived, designed and supervised the project. S.K., K.K., O.Q.H.Z., E.E.A. and H.S. performed experiments. K.K. and O.Q.H.Z. analyzed the data. K.K., O.Q.H.Z. and E.M.Ö. wrote the manuscript. S.K., E.E.A. and H.S. edited the manuscript.

## Competing interests
The authors declare no competing interests.
