## [Peer Review File · Nature Communications]

Stochastic gene expression and environmental stressors trigger variable somite segmentation phenotypesReviewer #1 (Remarks to the Author):

In this paper Keseroglu and Zinani et al., study possible reasons for why non-penetrant phenotypes arise in mutant genotypes. As a non-penetrant phenotype model, central segmentation clock players *her1* and *her7* are mutated leading to segmentation defects in a variety of percentages of mutated zebrafish offspring. It is well showcased how environmental factors such as hypoxia and temperature affect segment boundary formation and how the stochasticity in the expression of *her1* and *her7* lead to varying levels of segmentation defects. An interesting finding is the decrease in determination front regional amplitude of *her1* and *her7* signaling being strongly correlated with defective segment formation. This suggests a minimum threshold of *her* signaling requirement at the determination front to induce proper segment boundary formation.

The authors address a long-standing open question in the field of developmental biology, in which mutant animals show varying degrees of defective phenotypes. This would therefore not only have relevance for zebrafish embryos, but also for the study of other mutant animals, such as knockout mice (e.g. *Hes7* knockout mice).

While the paper is well written, I have a few suggestions to improve clarity and strengthen the conclusions made in this publication.

1. It is suggested that a low amplitude leads to a segmentation defect, however even in 'defective somites' the somite formation program is initiated. Some more discussion on how a low amplitude could lead to a less robust pattern or variable somite boundary formation would be helpful.
2. In Figure 2 it seems that many features of the wave dynamics are affected while the paper focusses solely on determination front amplitude. Other features of the altered wave dynamics could be the cause of the phenotype while these are not further explored.
3. In Figure 3 the embryos imaged are overexpression constructs, it would be good to check the expression levels of *her1* and *her7* compared to WT embryos.
4. In Figure 3 the amplitude reduction of the region could be due to desynchronization of the cells, to exclude this option single cell analysis of *her1* and *her7* should ideally be performed. This option should at least be discussed in this paper.
5. To strengthen the argument that hypoxia and temperature similarly increase phenotype penetrance by increasing stochasticity in gene expression, as is suggested in line 219-224, experiments to produce figure 2D and figure 3h could be repeated under hypoxic conditions or at lower temperatures.

Minor points

1. In Figure 2 a wholemount smFISH and the correlation between single *her1* and *her7* RNA molecules in single cells is shown. The correlation analysis is difficult to judge based on the images above so it would be nice to show either as supplement or as part of this figure some more information on what the single mRNA count data looks like (For example a magnification to show the spots in single cells). In addition, I assume N =fish pairings and n =embryos but the number of cells/mRNAs analyzed is unclear.
2. In figure 1B it is unclear to me when a somite boundary is considered defective and when it is considered WT. The boundary defect is clearer in 1E. A more explicit explanation of how defects are scored would support understanding of the main findings. I assume 1B shows a mild phenotype while 1E shows a more severe penetrance.
3. In figure 1C-D it is unclear whether the same data is used leading to an unclarity in the number of mutant embryos used in 1D.
4. In extended data figure 1 it would be nice to include an image and analysis of a non-affected embryo.
5. In Figure 3A-C it would be good to add the genotype and the mild, moderate and strong wording to the figure and include a WT embryo.

Reviewer #2 (Remarks to the Author):

In this paper, Keseroglu et al., use the segmentation clock of zebrafish to investigate penetrance and expressivity of tissue-level phenotypes. They combine in situ hybridization, time-lapse imaging and quantification of morphological defects. There are several interesting observations in this paper, but my concern is whether they indeed reveal new general insights about developmental variability, or are more interesting to specialists in the field. Furthermore, some of the results are not well-positioned in the context of the literature.

Points of concern

1. Line 67. b567 is a large deletion with many genes missing. The genetic studies that that proved which genes are responsible for the segmentation phenotype used mutations in only her1 and her7, these studies should be cited in addition: Llereas et al., 2018 and Zinani et al., 2021.
2. The results described in lines 80-87 have already been reported and quantified in Schröter et al., 2012. In Figure 4 and Fig S4. This part should be should be reported as "we confirmed". In this regard, the new ci302 allele matches the published results for hu2124, and the hu 2526 allele was previously published and analysed.
3. Fig 1e. It is hard to tell from these images where the boundaries are defective, and which side they are supposed to be on. Since this data is one of the main foundations of the paper, it would be good to have some much clearer images and annotation of the relevant defects.
4. Line 90 – It is hypothesized that due to stochastic gene expression in the Her-based clock, some groups of cells are oscillating while some other cell groups are not. But, if the cells had noisy single-cell clocks, would they not drift out of phase with each other and thereby lose the spatial group structure necessary for the formation of normal boundaries? Some form of local coordination between the cells may be necessary. There is an existing hypothesis in the literature that Delta-Notch coupling is involved in the variable expressivity of boundary formation Uriu et al., 2021. This paper analyses the left- and right-hand sides of embryos that de-synchronize and then re-synchronize, and shows that left- and right-hand sides are independent during resynchronization. Although the underlying mechanism is thought to be different to the proposed mechanism in this paper, the phenotype is very similar. This alternative hypothesis should be introduced and discussed.
5. Figure 1j. In the plot axis, please change defected boundary to number of defected boundaries, otherwise it can be interpreted as the ordinal boundary number.
6. Line 108-109. A similar low-temperature increase in the severity of the segmentation clock phenotype at 23.5 and 20 degrees has already been reported for the hes6 mutant and should be cited (Kawamura et al., 2005, Fig. 3; Schröter 2010, FigS1, Table S1).
7. Line 143. The stripe angle is different in WT, her1 mutants and her7 mutants, and a poor estimate of this angle would reduce the correlation. In a previous paper, the stripe angle along the axis was taken into account (Zinani et al., 2021), but it's not clear how this was addressed for the mutant data here. A poor estimate of the stripe angle would reduce the Pearson's correlation between her1 and her7.
8. Line 149. The authors state that the reduction of the gene expression correlation score in the mutants was "explaining their somite segmentation deficient phenotype", but it is not clear what is meant by this. The her1 and her7 mutants have defected boundaries in either the anterior or posterior of the axis, respectively. For the her1 mutant, at the developmental stage of the experiment shown in Figure 2, the defective boundaries are already formed, thus the correlation of the gene expression is in fact matched to normally forming boundaries. Thus, in contrast to the claim, the data do not explain the phenotype of the her1 mutant. If the data explain the her7 phenotype, then does the her1 phenotype have a different origin? Can the authors clarify what they mean?
9. Line 150-152. Where does the amplitude hypothesis come from? The subtitle is misleading, as

amplitude has not been shown to be required for segmentation, there is a correlation.

10. Figure 3d. As above, please change the axis to state "number of defected boundaries".

11. Line 163-164. In order to claim that the transgenes are functional, the the transgenes must be tested in isolation (her1-Venus in her1 mutant and her7-venus in her7 mutant). From the data presented here on posterior segmentation, it is possible that only the her7-Venus transgene is active. It seems to have less defects than the her7 mutants (8 at 26°C between somite 11 and 25), and this could well be due to the copy number of the transgene. How do the authors reach the conclusion that "successful segmentation can be attributed to reporter amplitudes" at this point?

12. Line 171-174. As presented, the link between reporter oscillation amplitude at the "determination front" and the formation of a given segment is not convincing. When looking at Fig. 3g, there is apparently only one defective segment, but there are 3 or 4 neighbouring low-amplitude cycles. When examining the traces of the medium and high expressivity cases in Extended Figure 1, there are extended intervals over which oscillations cannot be clearly identified. While this is an interesting line of investigation, and there does seem to be a correlation between a drop in normalized detrended intensity of the reporter signal and the probability of a defected boundary in the region, the claim that "an amplitude threshold of oscillations is required for segmentation" is not well supported. Either the claim should be altered, or more information is needed to understand this claim.

For example, it was not clear how the formation of defective segment boundaries was quantified. Are the somite boundaries scored from the time-lapse movies confirmed by in situ hybridization against *xirp2a*? Given the imaging set-up using a wide-field microscope and a 10x objective the depth of field may not distinguish between the left- and right-hand sides, how is this related to the side of an observed boundary defect? Why is the detrended normalized intensity used to assess the levels of Her proteins (a negative amount of a protein is not physical)? In terms of stochastic gene expression, what is the interpretation of lower signal intensity in a ROI that encompasses many (100s?) of cells? What is the correlation between a Her reporter signal at a location in the PSM that is not the determination front and a boundary, i.e. does the determination front give the best predictive power?

13. Line 396. (e) should be (h).

Reviewer #3 (Remarks to the Author):

Overall comment

In this study, Keseroglu et al describe that genetic defects for segmental clock genes stochastically fail to pass the amplitude threshold of oscillatory gene expression which results in the disruption of somitogenesis in zebrafish embryos. As phenotype expressivity often varies among genetically identical mutants, these findings may provide clues towards a better understanding of the incomplete penetrance and phenotypic variability of certain genetic diseases. Overall, the authors concisely provide data supporting their claims combined with a structured and neat description of their results. The manuscript is overall well-written and concise but would not suffer from a bit of polish. Furthermore, this is essentially an observational study since there are no attempts to unravel how stochastic gene expression actually relates to segment border formation or how lower temperature or hypoxia affect the penetrance of the clock gene mutations at the mechanistic and molecular level. The manuscript would hence strongly benefit from additional convincing molecular data and strong mechanistic evidence that would support and enhance the claims and conclusions of the authors. Nevertheless, I hope and am optimistic that the authors will be able to address my concerns listed below and that the updated version of this manuscript will satisfy the requirements necessary for publication and be of interest to the broad readership of Nature Communications.

Major concerns

1. In Fig.1 the authors show that environmental perturbations augment the strength of somite

segmentation in transgenic models. Considering the major claim and focus of this manuscript segmentation defects due to environmental factors should be displayed in a separate figure apart from Figs.1a-1h. Moreover, in order to show phenotypic variability of somite segmentation from diverse external stresses and stressors (including environmental factors), please also assess and validate whether and how embryotoxic chemicals (e.g., IPA-3 and SB225002; PMID: 29196645) augment phenotypic strength in her1 mutant models.

2. Please show the kymograph of the Venus signal in the her1; her7 transgenic models in Fig. 3 under the various temperature conditions and hypoxic condition shown in Fig. 1. Supporting data should be provided and assessed whether segmentation defects under various environmental stresses (as tested in Fig.1) have resulted from insufficient amplitude thresholds of clock oscillators.

3. Based on the guideline of Article submissions to "Nature Communications", the Discussion part should be succinct and may not include subheadings. Please concisely modify the Discussion section.

Minor issues

1. Correct the typo in line 24 of the abstract section; change "segmentation detects" to "segmentation defects".
2. In line 62, please replace the citation "Fig. 1a" with an appropriate reference as the sentence explains generally the role of her1 and her7 in genetic oscillations.
3. Please show and present the "defected boundary" of wildtype embryos as a negative control under various temperature conditions performed in Figs. 1i-1k.
4. Please show the anterior and posterior regions of the embryo in Fig. 2 for intuitive understanding of the figures.
5. Explain why did the authors quantify the Venus reporter amplitudes at 26oC which is lower than the normal culture temperature, 28oC in Fig. 3e.
6. Please show the biological and technical replications (n and N) in Extended Data Fig. 1.
7. Provide the detail information of "Fish care and embryo culture" in the Methods section.
8. Please specify and explain what the horizontal lines in Fig. 1d stand for and add an explanation into the corresponding figure legend.
9. In the legend of Fig.2, please specify that smFISH was performed to count her1 and her7 genes.
10. Please clarify how authors defined mild, moderate and strong phenotypes of segmentation in Fig.3 and Extended Data Fig. 1 based on the number of defective somites.
11. Please define "a.u." in figure legends and method section of the main text.

We thank all the Reviewers for their constructive feedback and complimentary remarks. We performed several new experiments, cited missing references, edited several sections in the text, and shortened the discussion section to bring our manuscript in line with the *Nature Communications* guidelines.

In this response letter document: the Reviewer comments will be in **black** font; our responses will be written in **blue** font, and any text called out from the manuscript will be in **green** font.

All new changes in the main manuscript are also highlighted in **green** font. Because Reviewer 3 asked us to present the environmental effects as a separate figure, we shifted the whole “Environmental factors can modify the strength of segmentation phenotypes” section to the end of the results section and moved the data to the new Figure 4. We did not change the color font of the old text that is shuffled, we only **green** colored the text newly added (due to new experimental data, citation or discussion). We also did not highlight the text removed from the shortened discussion section.

REVIEWER COMMENTS

Reviewer #1 (Remarks to the Author):

In this paper Keseroglu and Zinani et al., study possible reasons for why non-penetrant phenotypes arise in mutant genotypes. As a non-penetrant phenotype model, central segmentation clock players *her1* and *her7* are mutated leading to segmentation defects in a variety of percentages of mutated zebrafish offspring. It is well showcased how environmental factors such as hypoxia and temperature affect segment boundary formation and how the stochasticity in the expression of *her1* and *her7* lead to varying levels of segmentation defects. An interesting finding is the decrease in determination front regional amplitude of *her1* and *her7* signaling being strongly correlated with defective segment formation. This suggests a minimum threshold of *her* signaling requirement at the determination front to induce proper segment boundary formation.

The authors address a long-standing open question in the field of developmental biology, in which mutant animals show varying degrees of defective phenotypes. This would therefore not only have relevance for zebrafish embryos, but also for the study of other mutant animals, such as knockout mice (e.g. *Hes7* knockout mice).

While the paper is well written, I have a few suggestions to improve clarity and strengthen the conclusions made in this publication.

We thank the Reviewer for their complimentary remarks.

1. It is suggested that a low amplitude leads to a segmentation defect, however even in ‘defective somites’ the somite formation program is initiated. Some more discussion on how a low amplitude could lead to a less robust pattern or variable somite boundary formation would be helpful.

The clock provides regularity to the differentiation program by grouping cells into regular sized somite compartments. But partial somite boundaries form even in the double homozygous loss of function *her1;her7* (clockless) mutants for the following reason: Posteroanterior FGF/Wnt signaling gradients suppresses differentiation in pPSM and that process is derepressed and initiated in the aPSM independent of the clock. As cells differentiate in aPSM, they express certain adhesion molecules that later trigger cells to attach to each other and form regular boundaries in the presence of the clock and could lead to formation of random shaped clumps in clockless mutants. In the single *her1* and *her7* mutants, each embryo displays some failed and some successful boundaries.

Thank you for your suggestion. We have now inserted the following text (page 10, line 243): “In single *her1*^{ci302} or *her7*^{hu2526} mutants, the clock function stochastically fails during some cycles, resulting in low collective amplitudes among a population of cells. This failure then results in a less robust pattern and variable somite boundary formation as compared to wild-type embryos.”

2. In Figure 2 it seems that many features of the wave dynamics are affected while the paper focusses solely on determination front amplitude. Other features of the altered wave dynamics could be the cause of the phenotype while these are not further explored.

Earlier publications showed that the segmentation clock controls segment boundary formation solely at the determination front. Loss of clock waves (or expression) in aPSM could not stop segmentation (Giudicelli et al., PloS Biology, 2007). Similarly, mimicking the role of the clock can recreate segmentation only for cells located at the determination front; kinematic clock waves are irrelevant for somite boundary formation (Simsek et al., Nature, 2023). Therefore, we must analyze the clock at the determination front.

But our justification might be unclear in the original submission. We thank the Reviewer for bringing up this important issue. We have now inserted the following text to the manuscript (page 6, line 154): “Segmental commitment occurs in the middle of the PSM at a position called the determination front¹⁰, where the clock instructs segment boundaries^{22,25}. Loss of clock waves or expression in the aPSM could not stop segmentation²² and the kinematic clock waves are irrelevant for somite boundary formation²⁵.”

3. In Figure 3 the embryos imaged are overexpression constructs, it would be good to check the expression levels of *her1* and *her7* compared to WT embryos.

Our *her1* and *her7* probes cannot differentiate endogenous and exogenous genes from each other. Therefore, we performed smFISH experiments by using a *Venus* probe and counted the total number of *her1-Venus* plus *her7-Venus* RNAs (in the same genetic background). The total *Venus* RNA number is 64% of the total number of endogenous *her1* plus *her7* RNAs in wild-type embryos. Thus, the variable expressivity of the phenotype is not due to extreme overexpression of fusion reporter proteins.

We also would like to stress that protein copy numbers, in principle, do not matter for our experimental purpose because we are not making a relationship between protein numbers and segmentation rescue. We only wanted to (1) engineer a genetic background displaying variable phenotypic expressivity and (2) link the quantified oscillation amplitudes to the phenotype of each boundary (in the absence of confounding endogenous proteins). We also would like to stress that we are not claiming the *Venus* fusion proteins are equally functional as endogenous proteins. But they satisfy the two conditions written above and serve their purpose.

We have now inserted the following text to the manuscript (page 6, line 136 and 146): “To test the threshold hypothesis, we sought to quantify the amplitude of oscillations by imaging clock reporters in real-time in a genetic background displaying variable expressivity and containing only the reporters as the functional clock genes...” and “...Because endogenous clock proteins are absent in this genetic background, successful segmentation must be due to the reporter transgenes. We performed an smFISH experiment by using a probe against *Venus* and counted the total number of *her1-Venus* and *her7-Venus* RNAs in this genetic background. We found that the mean of transgene RNAs in mutant embryos was 36% lower than the mean of endogenous (*her1* plus *her7*) RNAs in wild-type embryos (Supplementary Fig. 2b), suggesting the reporter transgenes are at least partially functional.”

Supplementary Figure 2.

4. In Figure 3 the amplitude reduction of the region could be due to desynchronization of the cells, to exclude this option single cell analysis of *her1* and *her7* should ideally be performed. This option should at least be discussed in this paper.

We here report that the collective mean amplitude of a population of cells at the determination front stochastically decreases in single *her1* and *her7* mutants. We agree with the Reviewer that this decrease could be due to (1) an amplitude decrease in all cells or (2) desynchronization of a population of cells. In principle, either outcome would not change our conclusion: low collective amplitudes precede disrupted segmentation while higher ones precede successful ones.

Desynchronization of the cells is well studied in the somitogenesis field; it occurs due to weakened Notch signaling. Here, we do not weaken Notch signaling. That's why we do not favor the phenotype to emerge from desynchronization. We still agree with the Reviewer that we should discuss this alternative option and better explain our favored scenario. We have now inserted the following text to the manuscript (page 7, line 166): "The decrease in average amplitude of oscillations could be due to decreases of oscillation amplitudes in individual cells or desynchronization of oscillations among cells as seen when Delta/Notch signaling is weakened in the PSM^{19,27}. We do not favor the latter scenario because Her1 and Her7 repress transcription of *deltaC* ligand²² and therefore *her1* and *her7* clock mutants are expected to increase Delta/Notch signaling."

5. To strengthen the argument that hypoxia and temperature similarly increase phenotype penetrance by increasing stochasticity in gene expression, as is suggested in line 219-224, experiments to produce figure 2D and figure 3h could be repeated under hypoxic conditions or at lower temperatures.

We performed the requested real-time imaging experiments. The results confirmed that in all environmental conditions, successful boundaries are preceded by higher oscillation amplitudes. But, in less favorable environments, both the number of high amplitude oscillation cycles and successful segmentation proportionally decrease, agreeing with the prediction of the Reviewer 1 and strengthening our hypothesis.

Figure 4.

We have now inserted the following text to the manuscript (page 8, line 201): “We next quantified the amplitude of oscillations in the cells located at the determination front and the segmentation phenotype of those corresponding cells in *her1^{ci301}*; *her7^{hu2526}*; *her1-Venus^{+/+}*; *her7-Venus^{+/+}* embryos grown at a lower temperature of 21.5°C or grown at 26°C but under hypoxic conditions. We found that in both unfavorable conditions, the amplitude of oscillations preceding successful segmentation were higher than those preceding defective segmentation (Fig. 4e, f). Importantly, the number of cycles with high amplitudes and the number of successful segmentations are proportionally decreased in embryos grown in unfavorable environments as compared to 26°C controls (Fig. 4g, h).”

We have not performed the smFISH experiments under different environmental conditions. As we explain in more detail below (in response to one of the questions of Reviewer 2), the spatial Pearson correlation score is not sensitive enough to distinguish between *her1* and *her7* mutants. Thus, it cannot differentiate mutants grown under different environmental conditions as the difference in the number of defective boundaries will be milder than that of *her1* and *her7* mutants. Besides, the static smFISH experiments cannot report oscillation amplitudes.

Minor points

1. In Figure 2 a wholemount smFISH and the correlation between single *her1* and *her7* RNA molecules in single cells is shown. The correlation analysis is difficult to judge based on the images above so it would be nice to show either as supplement or as part of this figure some more information on what the single mRNA count data looks like (For example a magnification to show the spots in single cells). In addition, I assume N =fish pairings and n =embryos but the number of cells/mRNAs analyzed is unclear.

N is the number of independent experiments and n is the number of embryos used in Figure 2. We now report the number of cells and slices analyzed in each background in Methods.

We have now provided representative data of spatial *her1* and *her7* RNA profiles used for Pearson correlation analysis of each genotype (Supplementary Fig. 1).

2. In figure 1B it is unclear to me when a somite boundary is considered defective and when it is considered WT. The boundary defect is clearer in 1E. A more explicit explanation of how defects are scored would support understanding of the main findings. I assume 1B shows a mild phenotype while 1E shows a more severe penetrance.

Reviewer 2 also requested using better resolution images. We now replaced those images with newer ones. We inspected every boundary in an embryo by rotating it with a needle under microscope; thus we did not judge boundary phenotypes from microscopy images.

Boundaries are called defective only if they are fragmented or incomplete (as written in the methods).

That's correct, there are more broken boundaries in *her7* mutant (Fig. 1e) than in *her1* mutant (Fig. 1b).

3. In figure 1C-D it is unclear whether the same data is used leading to an unclarity in the number of mutant embryos used in 1D.

We have now edited the legend as follows (page 16, line 392): **c, d** Percentage of *her1^{ci302}* mutants displaying wild-type phenotype (striped) or segmentation defects (dotted) at 28°C (**c**, $n = 50$, $N = 15$), and number of defective boundaries per each side of *her1^{ci302}* mutants (**d**, $n = 100$, $N = 15$). The violin plot shows the median (thick black line) and quartiles (thin black lines).”

4. In extended data figure 1 it would be nice to include an image and analysis of a non-affected embryo.

In this genetic background, all embryos display segmentation defects but with variable expressivity. We provided the image of a wild-type embryos carrying the reporter transgenes (similar to the clock mutants). This figure is now called Supplementary Fig. 3.

Supplementary Figure 3.

5. In Figure 3A-C it would be good to add the genotype and the mild, moderate and strong wording to the figure and include a WT embryo.

We provided the image of a wild-type embryos carrying the reporter transgenes (similar to the clock mutants) into the Supplementary Fig. 3, so we omitted that in the main figure. We did not add classification to the figures because it might confuse some readers. We ordered the embryos in order of increasing number of defective boundaries. We inserted this in the legend as follows (page 24, line 504): "...ordered by their increasing number of defective boundaries (b through d)."

Reviewer #2 (Remarks to the Author):

In this paper, Keseroglu et al., use the segmentation clock of zebrafish to investigate penetrance and expressivity of tissue-level phenotypes. They combine in situ hybridization, time-lapse imaging and quantification of morphological defects. There are several interesting observations in this paper, but my concern is whether they indeed reveal new general insights about developmental variability, or are more interesting to specialists in the field. Furthermore, some of the results are not well-positioned in the context of the literature.

We thank Reviewer 2 for their complimentary remarks about our manuscript presenting "several interesting observations".

As the Reviewers 1 and 3 pointed out: our manuscript "address a long-standing open question in the field of developmental biology, in which mutant animals show varying degrees of defective phenotypes. This would therefore not only have relevance for zebrafish embryos, but also for the study of other mutant animals...", and the initial version of our manuscript showed that "the decrease in determination front regional amplitude of *her1* and *her7* signaling being strongly correlated with defective segment formation". Both Reviewer 1 and 3 also recommended us to test whether the amplitude threshold hypothesis holds true under unfavorable conditions (which affect the clock). We tested their prediction, and now we report additional data supporting our hypothesis and strengthening our conclusions.

In this resubmission, we have now properly cited the literature as requested by Reviewer 2, edited the text and added more data in this resubmission as requested by all Reviewers.

Points of concern

1. Line 67. b567 is a large deletion with many genes missing. The genetic studies that that proved which genes are responsible for the segmentation phenotype used mutations in only *her1* and *her7*, these studies should be cited in addition: Lleras et al., 2018 and Zinani et al., 2021.

Thank you for pointing to this omission. We have now additionally cited those two papers (on page 3, line 64): "#12 and #13".

2. The results described in lines 80-87 have already been reported and quantified in Schröter et al., 2012. In Figure 4 and Fig S4. This part should be should be reported as "we confirmed". In this regard, the new *ci302* allele matches the published results for *hu2124*, and the *hu 2526* allele was previously published and analysed.

We have now edited that section as follows (on page 4, line 85): "Similar to a previously published *her1*^{*hu2124*} line¹⁵, we found that sibling *her1*^{*ci302*} mutants displayed incomplete

penetrance (Fig. 1b): 15% of mutants successfully segmented all somites whereas 85% of them displayed defected boundaries (Fig. 1c, d, median defected boundary = 1). On the other hand, *her7^{hu2526}* mutants displayed complete penetrance of segmentation defects (Fig. 1e), confirming earlier results¹⁵. However, we additionally found that *her7^{hu2526}* embryos displayed variable expressivity (i.e., phenotypic strength among siblings); not all boundaries were broken in the posterior domain as intact and broken boundaries were intermingled (Fig. 1e).”

3. Fig 1e. It is hard to tell from these images where the boundaries are defective, and which side they are supposed to be on. Since this data is one of the main foundations of the paper, it would be good to have some much clearer images and annotation of the relevant defects.

We now replaced those images with newer ones.

4. Line 90 – It is hypothesized that due to stochastic gene expression in the Her-based clock, some groups of cells are oscillating while some other cell groups are not. But, if the cells had noisy single-cell clocks, would they not drift out of phase with each other and thereby lose the spatial group structure necessary for the formation of normal boundaries? Some form of local coordination between the cells may be necessary. There is an existing hypothesis in the literature that Delta-Notch coupling is involved in the variable expressivity of boundary formation Uriu et al., 2021. This paper analyses the left- and right-hand sides of embryos that desynchronize and then re-synchronize, and shows that left- and right-hand sides are independent during resynchronization. Although the underlying mechanism is thought to be different to the proposed mechanism in this paper, the phenotype is very similar. This alternative hypothesis should be introduced and discussed.

The desynchronization of the cells is well studied in the somitogenesis field. As the Reviewer refers to, desynchronization occurs when Notch signaling is weakened. *her1* and *her7* mutants do not weaken Notch signaling. That’s why we do not favor the phenotype to emerge from desynchronization. Nonetheless, we now cite to Uriu et al., 2021.

As we wrote above in response to the advice of Reviewer 1, we have now edited our manuscript as follows (on page 7, line 166): “The decrease in average amplitude of oscillations could be due to decreases of oscillation amplitudes in individual cells or desynchronization of oscillations among cells as seen when Delta/Notch signaling is weakened in the PSM^{19,27}. We do not favor the latter scenario because Her1 and Her7 repress transcription of *deltaC* ligand²² and therefore *her1* and *her7* clock mutants are expected to increase Delta/Notch signaling. Nonetheless, similar phenotypic variability between the left and right sides of embryos can also occur when Notch signaling is transiently pharmaceutically impaired²⁸.”

5. Figure 1j. In the plot axis, please change defected boundary to number of defected boundaries, otherwise it can be interpreted as the ordinal boundary number.

We implemented the requested change.

6. Line 108-109. A similar low-temperature increase in the severity of the segmentation clock phenotype at 23.5 and 20 degrees has already been reported for the *hes6* mutant and should be cited (Kawamura et al., 2005, Fig. 3; Schröter 2010, FigS1, Table S1).

We cited to these papers as follows (on page 7, line 177): “Furthermore, the severity of the segmentation defects increased in a different (*hes6*) mutant at low temperatures (Kawamura et al., 2005, Schröter 2010).”

7. Line 143. The stripe angle is different in WT, *her1* mutants and *her7* mutants, and a poor estimate of this angle would reduce the correlation. In a previous paper, the stripe angle along the axis was taken into account (Zinani et al., 2021), but it’s not clear how this was addressed

for the mutant data here. A poor estimate of the stripe angle would reduce the Pearson's correlation between *her1* and *her7*.

We reanalyzed this data by three different approaches: (1) All genotypes were analyzed with the same angle of wild-type. (2) All genotypes were analyzed with a fixed angle (angle = 90°, perpendicular to the A/P axis). (3) We used different angles for each genotype. Calculating stripe angles was particularly difficult in *her7* mutant. We used the angle demarcating the anterior most expression border as the angle throughout the PSM in this mutant. In the manuscript we decided to report the result of the third approach. We edited the methods as follows (on page 13, line 317): “We measured the angles of expression stripes in wild-type and *her1^{ci302}* mutants, fitted an equation to the data, and used them to divide tissue into separate slices, as previously described¹³. Measuring the stripe angles throughout the PSM in *her7^{hu2526}* mutants and *her1^{ci301}*; *her7^{hu2526}*; *her1-Venus^{+/-}*; *her7-Venus^{+/-}* transgene is difficult. Therefore, we used the mean of the angles measured at the anterior-most expression domain as a fixed angle for slicing the PSM in these mutants.”

(Figure not shown in the manuscript. This will be published as part of the peer review document if our manuscript is accepted)

In all three approaches, we obtained similar results. *her1* and *her7* mutants have lower expression correlation scores as compared to wild-type embryos. We agree with the Reviewer that the Pearson correlation score has a limit of resolution. It is decreased in both mutants as compared to wild-type embryos. It is understandable that both mutants lack some critical repressor dimers which can lower the correlation score in both mutants. But, as the Reviewer alludes to, this metric lacks the resolution to separate *her1* from *her7*. Hence, we edited the relevant section as follows (on page 5, line 126): “We found that the correlation score significantly decreased in both *her1^{ci302}* and *her7^{hu2526}* mutants as compared to wild-type embryos (Fig. 2d; Supplementary Fig. 1). But the Pearson correlation score could not differentiate between *her1^{ci302}* and *her7^{hu2526}* mutants; this could either be due to this metric reaching its resolution limit in *her1^{ci302}* mutant or the difficulty of precisely measuring the expression stripe angles in the clock mutants.”

8. Line 149. The authors state that the reduction of the gene expression correlation score in the mutants was “explaining their somite segmentation deficient phenotype”, but it is not clear what is meant by this. The *her1* and *her7* mutants have defected boundaries in either the anterior or posterior of the axis, respectively. For the *her1* mutant, at the developmental stage of the experiment shown in Figure 2, the defective boundaries are already formed, thus the correlation

of the gene expression is in fact matched to normally forming boundaries. Thus, in contrast to the claim, the data do not explain the phenotype of the *her1* mutant. If the data explain the *her7* phenotype, then does the *her1* phenotype have a different origin? Can the authors clarify what they mean?

We agree with the Reviewer, therefore we removed the old text, and instead inserted the green text written above in response to the previous comment. The Pearson correlation catches perfect stripe alignment in wild-type embryos, but the score decreases as soon as some stripe mismatch occurs in *her1* mutant. What's important here is that the correlation score is weaker in both mutants as compared to wild-type embryos. Therefore, these mutants both display segmentation defects in that region (much fewer in *her1* than *her7* mutants)

9. Line 150-152. Where does the amplitude hypothesis come from? The subtitle is misleading, as amplitude has not been shown to be required for segmentation, there is a correlation.

We agree with the Reviewer. We changed the subheading as follows (on page 6, line 132): "Successful segmentation is preceded by high amplitude clock oscillations".

10. Figure 3d. As above, please change the axis to state "number of defected boundaries".

We implemented the requested change.

11. Line 163-164. In order to claim that the transgenes are functional, the the transgenes must be tested in isolation (*her1*-Venus in *her1* mutant and *her7*-venus in *her7* mutant). From the data presented here on posterior segmentation, it is possible that only the *her7*-Venus transgene is active. It seems to have less defects than the *her7* mutants (8 at 26°C between somite 11 and 25), and this could well be due to the copy number of the transgene. How do the authors reach the conclusion that "successful segmentation can be attributed to reporter amplitudes" at this point?

This issue was also brought up by Reviewer 1. As reported above, we now counted the RNA numbers. The total transgenic RNA numbers are less than that of endogenous genes in wild-type embryos. The results suggest that we don't have too many copy numbers (as seen in other published lines in the field). The fact that we see less defects than in *her7* mutants means that both transgenes are partially functional.

As discussed above, for our purposes, both transgenes do not need to be functional (or equally functional). One reporter being functional and the other being a pure reporter would also be adequate. The only reason we generated this line is to obtain a genetic model displaying variable expressivity (and the clock protein(s) are quantifiable). We edited the relevant text as discussed above.

12. Line 171-174. As presented, the link between reporter oscillation amplitude at the "determination front" and the formation of a given segment is not convincing. When looking at Fig. 3g, there is apparently only one defective segment, but there are 3 or 4 neighbouring low-amplitude cycles. When examining the traces of the medium and high expressivity cases in Extended Figure 1, there are extended intervals over which oscillations cannot be clearly identified. While this is an interesting line of investigation, and there does seem to be a correlation between a drop in normalized detrended intensity of the reporter signal and the probability of a defected boundary in the region, the claim that "an amplitude threshold of oscillations is required for segmentation" is not well supported. Either the claim should be altered, or more information is needed to understand this claim.

As the Reviewer acknowledges, there is a strong correlation between higher amplitudes of oscillations and segmentation success (Fig. 3h). Of course, there is variability among individual data (or oscillation cycles). This is a universal fact seen in every study. That's why the

distributions of two groups overlap in panel Fig. 3h; part of this overlap could also be due to our experimental error or resolution limit.

Nonetheless, we agree with the Reviewer that one of our subheadings must be toned down. We changed it to: “Successful segmentation is preceded by high amplitude clock oscillations”. Now, we are not making any overstatement in the manuscript or claiming that we have proven a threshold is needed. We are simply providing a powerful correlation result and proposing a new hypothesis in the field. Certainly, several other correlative studies were previously published in our field proposing new inspiring hypotheses (such as Somites without a clock, Doppler effect, phase shift models). We believe our study is similar to those earlier studies and will lead to new avenues, both in the field and outside as well.

We now also provide additional data, as requested by Reviewers 1 and 3, showing that the strong correlation we previously obtained in favorable conditions is now extendable to unfavorable environments (i.e. low temperature or hypoxic environment, Fig. 4e, f).

For example, it was not clear how the formation of defective segment boundaries was quantified. Are the somite boundaries scored from the time-lapse movies confirmed by in situ hybridization against *xirp2a*? Given the imaging set-up using a wide-field microscope and a 10x objective the depth of field may not distinguish between the left- and right-hand sides, how is this related to the side of an observed boundary defect? Why is the detrended normalized intensity used to assess the levels of Her proteins (a negative amount of a protein is not physical)? In terms of stochastic gene expression, what is the interpretation of lower signal intensity in a ROI that encompasses many (100s?) of cells? What is the correlation between a Her reporter signal at a location in the PSM that is not the determination front and a boundary, i.e. does the determination front give the best predictive power?

We did not confirm boundaries by *xirp2*. Boundaries are scored by going over brightfield sections. We now added this missing information to the Methods (on page 14 line 340).

There could be some signal leakage between the two sides. This leakage will increase the technical noise in data. This technical noise could also partially explain why the amplitudes of two populations (defective and intact boundaries) partially overlap. What’s important here is that we don’t introduce any systematic biases to favor an outcome (one population over another).

We have now eliminated detrending in our analysis. As expected, it did not change the conclusions. But the profile plots will now be less confusing for readers.

Our interpretation is that when the collective amplitude of clock oscillations at the determination front is low, which occurs stochastically in single clock gene mutants, segmental determination fails.

The concept of the determination front is well established in the field of somitogenesis. Therefore, we ought to do the analysis at this location; this is not our subjective choice. Nonetheless, we calculated the amplitudes at two other locations (S-VII and S0) as requested. The results showed that the amplitudes of oscillations preceding successful segmentation are higher than those preceding failed ones in all three locations. The difference of amplitudes among the two groups are lowest at the posterior most PSM (S-VII). The difference at the determination front is slightly higher than that of S0, which is higher than that of the posterior most PSM (S-VII). We now report this figure as “Supplementary Figure 4”.

Supplementary Figure 4. The correlations of the clock amplitudes (measured at different locations) with segmentation success. a-c Cartoons show kymographs using PSM lengths between 5 and 26 somite stage as reported in ²⁶. Amplitudes are calculated along the S-VII (a, orange line, posterior PSM), S-III (b, blue line, determination front), and S0 positions (c, purple line, anterior PSM). d-f The amplitudes of oscillations preceding successful segmentation are higher than those preceding failed ones in all three locations, whereas the difference at the determination front is slightly better (median difference: 34.0, $n_{\text{intact}} = 283$, $n_{\text{defected}} = 328$, $N = 3$) than that of anterior PSM (median difference: 28.5, $n_{\text{intact}} = 248$, $n_{\text{defected}} = 365$, $N = 3$), which is better than that of the posterior PSM (median difference: 20.5, $n_{\text{intact}} = 230$, $n_{\text{defected}} = 214$, $N = 3$). ** $P = 0.0080$ (posterior), **** $P < 0.0001$ (determination front), **** $P < 0.0001$ (anterior). The whisker plot shows the median (line), quartiles (box), as well as the 10th and 90th percentiles (whiskers). n is the number of boundaries; N is the number of independent experiments.

13. Line 396. (e) should be (h).

We implemented the requested change. Thanks for catching this typo.

Reviewer #3 (Remarks to the Author):

Overall comment

In this study, Keseroglu et al describe that genetic defects for segmental clock genes stochastically fail to pass the amplitude threshold of oscillatory gene expression which results in the disruption of somitogenesis in zebrafish embryos. As phenotype expressivity often varies among genetically identical mutants, these findings may provide clues towards a better understanding of the incomplete penetrance and phenotypic variability of certain genetic diseases. Overall, the authors concisely provide data supporting their claims combined with a structured and neat description of their results. The manuscript is overall well-written and concise but would not suffer from a bit of polish. Furthermore, this is essentially an observational study since there are no attempts to unravel how stochastic gene expression actually relates to segment border formation or how lower temperature or hypoxia affect the penetrance of the clock gene mutations at the mechanistic and molecular level. The manuscript would hence strongly benefit from additional convincing molecular data and strong mechanistic evidence that would support and enhance the claims and conclusions of the authors. Nevertheless, I hope and am optimistic that the authors will be able to address my concerns listed below and that the updated version of this manuscript will satisfy the requirements necessary for publication and be of interest to the broad readership of Nature Communications.

We thank the Reviewer for their complimentary remarks. We now incorporated results from additional requested experiments, strengthening our conclusions and supporting our hypothesis.

Major concerns

1. In Fig.1 the authors show that environmental perturbations augment the strength of somite segmentation in transgenic models. Considering the major claim and focus of this manuscript segmentation defects due to environmental factors should be displayed in a separate figure apart from Figs.1a-1h. Moreover, in order to show phenotypic variability of somite segmentation from diverse external stresses and stressors (including environmental factors), please also assess and validate whether and how embryotoxic chemicals (e.g., IPA-3 and SB225002; PMID: 29196645) augment phenotypic strength in *her1* mutant models.

We split the environmental effect data from Fig. 1 into Fig. 1 and Fig. 4 as requested. We performed the requested additional experiments. All these changes caused reshuffling of the order of figures and some text in the manuscript.

We found that these two previously reported chemicals cause segmentation defects in both wild-type and *her1* mutants. Our conclusion is that a low dose only induced defects in *her1* mutant, while defects are found in both backgrounds at a higher dose. Because the referred previous study (PMID: 29196645) showed that these chemicals do not disturb the segmentation clock, we assume that they function either in parallel to the clock or act on processes occurring at later time points. Further research is needed to find their molecular targets. Since these chemicals are studied by an eminent group in our field, we assume they are chasing after these molecules.

Here, we have reported our results as follows (on page 9, line 209): "Besides natural environmental factors, like temperature and hypoxia, other embryotoxic chemicals were previously shown to worsen segmentation defects in zebrafish³². We confirmed that two of those chemicals (IPA-3 and SB225002) cause segmentation defects at moderate (3 μ M) (Fig. 4i, j) but not low (1 μ M) (Supplementary Fig. 5) doses in wild-type embryos. Since previous work showed that these two chemicals do not directly affect the segmentation clock³², these drugs might impair processes occurring after segmental commitment. Future studies may investigate the specific molecular pathways that are assaulted by these chemicals."

Figure 4.

Supplementary Figure 5.

2. Please show the kymograph of the Venus signal in the *her1*; *her7* transgenic models in Fig. 3 under the various temperature conditions and hypoxic condition shown in Fig. 1. Supporting data should be provided and assessed whether segmentation defects under various environmental stresses (as tested in Fig.1) have resulted from insufficient amplitude thresholds of clock oscillators.

Because Reviewer 1 also requested these experiments, we reported the data above. We thank both Reviewers for their constructive suggestions; these results strengthen our conclusions.

3. Based on the guideline of Article submissions to “Nature Communications”, the Discussion part should be succinct and may not include subheadings. Please concisely modify the Discussion section.

We removed the subheadings and shortened the discussion.

Minor issues

1. Correct the typo in line 24 of the abstract section; change “segmentation detects” to

“segmentation defects”.

We implemented the requested change. Thanks for catching this typo.

2. In line 62, please replace the citation “Fig. 1a” with an appropriate reference as the sentence explains generally the role of *her1* and *her7* in genetic oscillations.

We implemented the requested change and cited to Refs #11-13.

3. Please show and present the “defected boundary” of wildtype embryos as a negative control under various temperature conditions performed in Figs. 1i-1k.

No defected boundary was detected in all WT embryos (n:100) raised at 21°C, 23°C, and 28°C. We reported the data in Fig. 4a.

4. Please show the anterior and posterior regions of the embryo in Fig. 2 for intuitive understanding of the figures.

We highlighted the anterior and posterior directions for the readers.

5. Explain why did the authors quantify the Venus reporter amplitudes at 26°C which is lower than the normal culture temperature, 28°C in Fig. 3e.

Our sole goal was to find a genotype at a condition mimicking variable expressivity. This temperature served this purpose. There was no other reason. Several fish aquariums are maintained at 26°C-27°C.

6. Please show the biological and technical replications (n and N) in Extended Data Fig. 1.

This figure is now called Supplementary Fig. 3. We didn't group embryos based on their segmentation phenotype. Therefore, the biological and technical replications are the same as in Figure 3. Nonetheless, Fig. 3d shows the number of defected boundaries between 11th and 25th somite boundaries, which shows the variety of phenotypic strengths.

7. Provide the detail information of “Fish care and embryo culture” in the Methods section.

We now provided those details on page 11, line 261.

8. Please specify and explain what the horizontal lines in Fig. 1d stand for and add an explanation into the corresponding figure legend.

We now provided those details in the legend.

9. In the legend of Fig.2, please specify that smFISH was performed to count *her1* and *her7* genes.

We edited the legend of Fig. 2 as: “...smFISH was performed at 12-14 somite stage to count *her1* (cyan) and *her7* (red) transcription...”

10. Please clarify how authors defined mild, moderate and strong phenotypes of segmentation in Fig.3 and Extended Data Fig. 1 based on the number of defective somites.

We did not use a particular quantitative metric. That's why we did not report any pie chart showing how many embryos fall into each category. This figure's purpose was to show that a variety of phenotypic strengths are seen, even in a clutch of sibling embryos grown in identical conditions. To clarify this issue, we have now edited the legend text as:

“...Representative embryos are ordered by their increasing number of defective somites (a though c).”

11. Please define “a.u.” in figure legends and method section of the main text.

We edited the text in the legend of Fig. 3 as: “...Smoothened line profile of Venus signal along determination front position (arbitrary unites, a.u.)...” and in the methods: “...the raw intensity (arbitrary unites, a.u.) was smoothened...”

Reviewer #1 (Remarks to the Author):

My comments on the manuscript entitled "Stochastic gene expression and environmental stressors trigger variable somite segmentation phenotypes" have been addressed sufficiently.

As the authors confirm, the connection between oscillation amplitude and probability of somite defects is a correlation. This should be stated clearly throughout the manuscript.

Reviewer #3 (Remarks to the Author):

Overall comment

We are mostly satisfied with the authors replies to our comments and suggestions. They added clarifications and extra data where needed. We would still appreciate if the authors could respond to the subsequent comments, as it would contribute to a more comprehensive version of the manuscript.

1. Please show the kymographs of the temperature and hypoxia oscillation amplitude measurement experiments for the sake of completion, since the data is now found in Figure 4.
2. Please provide the bright-field images for all the graphs found in Figure 4, similar to the presentation in Figure 3a-c, where the quantification was based on the "# of Defected Boundary."

Minor points

1. Line 31. Changed "adaptable" to "adapted".
2. Line 215. Changed "assaulted" to "affected".
3. Provide a definition for "ns," indicating it as "not significant," and include this term in the respective figure legends for all graphs where applicable.

Congratulations to the authors for this nice and interesting piece of work.

We thank the Reviewers for their complimentary remarks. They both only had minor comments. In this response letter document: the Reviewer comments will be in **black** font; our responses will be written in **blue** font, and any text called out from the manuscript will be in **green** font. All new changes in the main manuscript are also highlighted in **green** font; these changes are due to the requests of the two reviewers or the editor.

REVIEWER COMMENTS

Reviewer #1 (Remarks to the Author):

My comments on the manuscript entitled "Stochastic gene expression and environmental stressors trigger variable somite segmentation phenotypes" have been addressed sufficiently. Thank you.

As the authors confirm, the connection between oscillation amplitude and probability of somite defects is a correlation. This should be stated clearly throughout the manuscript.

We have now explicitly noted this as a correlation in the summary paragraph of the introduction with: "Through live imaging of the segmentation clock reporters, we further show that segmentation success correlates with the collective oscillation amplitude." To further address this, we titled figure legend 5 as "Figure 5. Segmentation clock amplitudes correlate with segmentation phenotypes affected by environmental factors." We also edited the relevant sentence in the results section on lines 168-169 as: "This result shows that higher oscillation amplitude correlates with successful somite segmentation."

Reviewer #3 (Remarks to the Author):

Overall comment

We are mostly satisfied with the authors replies to our comments and suggestions. They added clarifications and extra data where needed. We would still appreciate if the authors could respond to the subsequent comments, as it would contribute to a more comprehensive version of the manuscript.

1. Please show the kymographs of the temperature and hypoxia oscillation amplitude measurement experiments for the sake of completion, since the data is now found in Figure 4. We added the requested kymographs of low temperature and hypoxia data. As these additions grew the figure (and the legend would be more than 350 words), we split the old figure 4 into 3 figures (Figures 4-6). The requested kymographs are provided in Fig. 5b,e. Although not requested, for the sake of completion, we also provided their line plots in panels Fig. 5c, f.

2. Please provide the bright-field images for all the graphs found in Figure 4, similar to the presentation in Figure 3a-c, where the quantification was based on the "# of Defected Boundary."

We added the requested brightfield images in the newly created Fig. 5a, d.

Minor points

1. Line 31. Changed "adaptable" to "adapted".

We edited this sentence slightly differently.

2. Line 215. Changed "assaulted" to "affected".

We made this change.

3. Provide a definition for "ns," indicating it as "not significant," and include this term in the respective figure legends for all graphs where applicable.

We defined "ns" as "not significant" in respective figure legends.

Congratulations to the authors for this nice and interesting piece of work.

Thank you.